# Berberine-Based Carbon Quantum Dots Improve Intestinal Barrier Injury and Alleviate Oxidative Stress in C57BL/6 Mice with 5-Fluorouracil-Induced Intestinal Mucositis by Enhancing Gut-Derived Short-Chain Fatty Acids Contents

**DOI:** 10.3390/molecules28052148

**Published:** 2023-02-24

**Authors:** Liang Wu, Yue Xi, Man Yan, Chang Sun, Jiajun Tan, Jiayuan He, Haitao Li, Dongxu Wang

**Affiliations:** 1Department of Laboratory Medicine, School of Medicine, Jiangsu University, Zhenjiang 212013, China; 2Medical Laboratory Department, Huai’an Second People’s Hospital, Huai’an 223022, China; 3Health Testing Center, Zhenjiang Center for Disease Control and Prevention, Zhenjiang 212002, China; 4Energy Research Institute, Jiangsu University, Zhenjiang 212013, China; 5School of Grain Science and Technology, Jiangsu University of Science and Technology, Zhenjiang 212100, China

**Keywords:** berberine, carbon quantum dots, intestinal mucositis, intestinal mucosal barrier, short-chain fatty acids

## Abstract

This study aims to evaluate the effect of berberine-based carbon quantum dots (Ber-CDs) on improving 5-fluorouracil (5-FU)-induced intestinal mucositis in C57BL/6 mice, and explored the mechanisms behind this effect. Thirty-two C57BL/6 mice were divided into four groups: normal control (NC), 5-FU-induced intestinal mucositis model (5-FU), 5-FU + Ber-CDs intervention (Ber-CDs), and 5-FU + native berberine intervention (Con-CDs). The Ber-CDs improved body weight loss in 5-FU-induced mice with intestinal mucositis compared to the 5-FU group. The expressions of IL-1β and NLRP3 in spleen and serum in Ber-CDs and Con-Ber groups were significantly lower than those in the 5-FU group, and the decrease was more significant in the Ber-CDs group. The expressions of IgA and IL-10 in the Ber-CDs and Con-Ber groups were higher than those in the 5-FU group, but the up-regulation was more significant in the Ber-CDs group. Compared with the 5-FU group, the relative contents of Bifidobacterium, Lactobacillus and the three main SCFAs in the colon contents were significantly increased the Ber-CDs and Con-Ber groups. Compared with the Con-Ber group, the concentrations of the three main short-chain fatty acids in the Ber-CDs group were significantly increased. The expressions of Occludin and ZO-1 in intestinal mucosa in the Ber-CDs and Con-Ber groups were higher than those in the 5-FU group, and the expressions of Occludin and ZO-1 in the Ber-CDs group were more higher than that in the Con-Ber group. In addition, compared with the 5-FU group, the damage of intestinal mucosa tissue in the Ber-CDs and Con-Ber groups were recovered. In conclusion, berberine can attenuate intestinal barrier injury and oxidative stress in mice to mitigate 5-fluorouracil-induced intestinal mucositis, moreover, the above effects of Ber-CDs were more significant than those of native berberine. These results suggest that Ber-CDs may be a highly effective substitute for natural berberine.

## 1. Introduction

Chemotherapy-induced intestinal mucositis (CIM), which is the most common adverse reaction in patients with cancer receiving chemotherapy, manifests as gastrointestinal problems, such as nausea, vomiting, diarrhea, appetite loss, indigestion and malabsorption [1]. CIM increases the pain of patients with cancer and often causes them to discontinue chemotherapy, thus reducing the survival rate of these patients. However, effective prevention and treatment strategies for CIM remain to be devised [2]. CIM has an estimated incidence rate of 40%, and 90% of all cases involve the use of 5-fluorouracil (5-FU) and methotrexate [3]. 5-FU is used to treat various gastrointestinal tract tumors, including colon, stomach, and esophageal tumors [4]. Few treatment options are available for 5-FU-induced CIM, thus necessitating studies aimed at developing effective drugs for this condition.

The maintenance of gut homeostasis requires high levels of energy, which is supplied primarily by mitochondria [5], which are not only the primary source of reactive oxygen species (ROS) under oxidative stress but also a key drug target [6]. Under oxidative stress conditions, damaged mitochondria can produce 10 times more ROS than normal mitochondria can, thus aggravating the intestinal histologic damage [7]. In the absence of an adequate antioxidant system, excess levels of ROS can compromise epithelial cell integrity and intestinal barrier function by reducing the number of tight junctions and the quality of cells [8]. ROS and other free radicals can disrupt cell functions by interfering with transcription factors and redox-sensitive signaling pathways. The nucleotide-binding oligomerization domain-, leucine-rich repeat-, and pyrin domain-containing protein 3 (NLRP3) inflammasome is a tripartite protein that regulates the oxidative status and inflammation of cells [9]. ROS are involved in the development of intestinal injury; therefore, cells must effectively alleviate oxidative stress, which may be associated with increased intestinal permeability and short-chain fatty acids (SCFAs) levels, and reduced epithelial apoptosis.

Berberine (Ber) is a quaternary ammonium alkaloid found in 4 families and 10 genera of plants (e.g., *Berberis* aristate); high levels of Ber have been detected in *Coptis coptidis* and *Phellodendrus chinensis* [10]. Ber exerts strong anti-inflammatory effects, but the exact underlying mechanism remains to be fully understood [11]. Carbon quantum dots (CDs) are surface-modified carbon nanoparticles, which composed a carbon core and an external carbon shell [12]. CDs exhibit high fluorescence stability, photobleaching resistance, broad and continuous fluorescence emission upon excitation, good water solubility, and rapid migration in cells and are thus considered to be ideal biomedical materials [13,14]. Liu et al. [15] designed a type of nitrogen-doped CD and demonstrated its biological activity in inhibiting β-amyloid aggregation. The use of traditional Chinese medicine in the preparation of CDs may enhance drug solubility and ensure sustained drug release in vivo, thus substantially increasing the effectiveness of these CD-based drugs [16]. The main disadvantage of Ber is its poor solubility, which can be markedly enhanced by using a different dosage form [17].

In this study, we used a mouse model (C57BL/6) of 5-FU-induced intestinal mucositis to investigate the effects of berberine-based carbon quantum dots (Ber-CDs) on intestinal barrier injury, mitochondrial function, and inflammation, and to elucidate the underlying molecular mechanisms.

## 2. Results

### 2.1. Preparation and Characterization of Ber-CDs

High-resolution images obtained through transmission electron microscopy showed that the carbon particles were complete crystals without any internal defects, and the Ber-CDs were spherical monodisperse particles with a diameters of about approximately 2–5 nm (Figure 1A). Figure 1B presents the emission spectrum of Ber-CDs obtained under the condition of a fixed excitation light wavelength. The fluorescence emitted by the samples at different wavelengths was measured. A graph was drawn using the wavelength of Ber-CD-emitted fluorescence as the *x*-axis and the intensity (relative fluorescence units, RFU) as the *y*-axis. The maximum emission wavelength of Ber-CDs at an excitation wavelength of 380 nm was 518 nm. With a fixed emission wavelength of 450 nm, the maximum excitation wavelength of Ber-CDs was 380 nm (Figure 1C). In addition, the performance of synthesized Ber-CDs with different preservation times were measured by fluorescence spectrometer as shown in Appendix A, and these results suggest that the synthesized Ber-CDs has good stable performance.

### 2.2. The Body Weight Changes of Experimental Mice

Figure 2 presents the body weight of each group of mice during the experimental period. The normal control (NC) group exhibited a continual increase in body weight throughout the experimental period, whereas the 5-FU, Ber-CD, and native Ber (Con-Ber) groups exhibited significant reductions in body weight 3 days after receiving 5-FU injection (intraperitoneal); in these three group, body weight continued to decrease during 5-FU treatment. On Day 8, the 5-FU group had the lowest body weight, followed by the Con-Ber group; however, no significant difference was observed between the two groups (*p* > 0.05). The body weight of the Ber-CD group was significantly higher than that of the 5-FU and Con-Ber groups (*p* < 0.05).

### 2.3. Expressions Levels of Inflammatory Factors in the Spleen and Serum of Mice

Figure 3 shows the changes in the expression levels of NLRP3, interleukin (IL)-10, and IL-1β, which indicate the level of inflammatory response in mice, and that of immunoglobulin (Ig)A, which indicates the degree of mucosal immunity in mice. Compared with the findings in 5-FU group, the expression level of NLRP3 in the spleen significantly decreased and that of IL-10 significantly increased in the Ber-CD and Con-Ber groups (*p* < 0.05). Compared with the findings in the Con-Ber group, the expression level of NLRP3 significantly decreased and that of IL-10 significantly increased in the Ber-CD group (*p* < 0.05). Enzyme-linked immunosorbent assay (ELISA) revealed that compared with the findings for the 5-FU group, the serum level of IL-1β significantly decreased but that of IgA significantly increased in the Ber-CD and Con-Ber groups (*p* < 0.05). Compared with the findings in the Con-Ber group, the serum level of IL-1β significantly decreased and that of IgA significantly increased in the Ber-CD group (*p* < 0.05).

### 2.4. Morphology of the Intestinal Mucosa and Expression of Tight Junction Proteins

Figure 4A depicts the mouse intestinal mucosa stained with hematoxylin–eosin (HE). In the NC group, the intestinal mucosa was intact and undamaged, the intestinal gland body was arranged regularly, and the mucosal lamina propria was unchanged. In the 5-FU group, the intestinal mucosa was severely necrotic; mucosal atrophy, villi shedding, and mutilated glands were noted; and crypts disappeared. The aforementioned injuries were improved to a certain extent in the Ber-CDs and Con-Ber groups, wherein the small intestinal mucosal epithelium was reduced, glands were relatively neat, and mucosal layer necrosis was inhibited. Figure 4B presents the results of immunohistochemical analyses performed to evaluate the expression levels of the mucosal barrier integrity proteins occludin and zonula occludens (ZO)-1 in the intestinal mucosal tissues of the mice. Compared with the findings for the NC group, the expression levels of ZO-1 and occludin in the small intestine significantly decreased in the 5-FU and Con-Ber groups (*p* < 0.05). Compared with the findings for the 5-FU group, the expression levels of both ZO-1 and occludin significantly increased in the Ber-CD group (*p* < 0.05) but only that of occludin significantly increased in the Con-Ber group (*p* < 0.05).

### 2.5. Relative Abundances of Intestinal Bacteria and Levels of SCFAs in Mouse Feces

The relative abundances of four important bacteria in the colon contents of the mice were evaluated through quantitative reverse-transcription polymerase chain reaction (qRT-PCR). The results are presented in Figure 5A. Compared with the findings for the NC group, the relative abundances of *Bifidobacterium* and *Lactobacillus* significantly decreased in the 5-FU group (*p* < 0.05); however, no significant between-group difference was noted in the relative abundances of *Escherichia coli* and *Enterococcus* (*p* > 0.05). Compared with the findings for the 5-FU group, the relative abundances of *Bifidobacterium* and *Lactobacillus* significantly increased in the Ber-CD and Con-Ber groups (*p* < 0.05). However, we noted no significant between-group difference in the relative abundances of *E. coli* and *Enterococcus* (*p* > 0.05).

The levels of three main SCFAs in colon contents of the mice were measured through liquid chromatography (LC) with tandem mass spectrometry (MS/MS). The results are presented in Figure 5B. Compared with the findings for the NC group, the levels of these SCFAs significantly decreased in the 5-FU group (*p* < 0.05). Compared with the findings for the 5-FU group, the levels of these SCFAs significantly increased in the Ber-CD and Con-Ber groups. The levels of propionic and butyric acids were significantly lower in the NC group than in the other groups (*p* < 0.05). Compared with the findings for the Con-Ber group, the levels of the three main SCFAs significantly increased in the Ber-CD group (*p* < 0.05).

### 2.6. Levels of Plasma Endotoxin, Superoxide Dismutase and Malondialdehyde Concentrations in Mice

Compared with the findings for the NC group, the level of endotoxin significantly increased in the 5-FU, Ber-CD, and Con-Ber groups (*p* < 0.05; Figure 6A); however, no significant differences were noted between these three treatment groups (*p* > 0.05). Compared with the findings for the NC group, the level of superoxide dismutase (SOD) was significantly decreased only in the 5-FU group (*p* < 0.05; Figure 6B); compared with the level of SOD in the Con-Ber group, that in the Ber-CD group was increased significantly (*p* < 0.05). Compared with the findings for the NC group, the level of malondialdehyde (MDA) was significantly increased in the 5-FU and Con-Ber groups (*p* < 0.05; Figure 6C), but no significant difference was observed among the other groups (*p* > 0.05).

## 3. Discussion

Recent studies on CDs, a new nanoscale luminescent carrier, have focused primarily on optimizing their preparation method (e.g., using traditional Chinese medicine) and expanding their range of applications. CDs exhibit good solubility and dispersion properties because of their small size and thus may markedly improve the efficacy of traditional Chinese medicine [12]. CDs with a particle size of < 10 nm can be formed by subjecting traditional Chinese medicine to certain physical and chemical reactions. This indicates that compared with traditional Chinese medicine, CD-based traditional Chinese medicine exhibits enhanced photoluminescence, reduced toxicity, and improved water solubility and biological compatibility. The use of traditional Chinese medicine in the preparation of CDs may resolve the insolubility-related problems of some Chinese medicines and even confer new biological activities [18]. Therefore, the use of traditional Chinese medicine in the preparation of CDs with fluorescence stability and pharmacological activity may increase drug efficacy and expand drug applications.

Ber is a quaternary ammonium alkaloid isolated from the traditional Chinese medicine *Coptis coptidis*. This compound exhibits a wide range of pharmacological activities, including anti-inflammatory, anti-infection, and anti-tumor activities. Few studies have explored the effects of Ber on CIM [19]. Ber and its hydrochloride or sulfate are insoluble in water, which substantially limits their pharmacological use [20]. In our study, CDs were added to Ber to increase the solubility of native Ber, and excellent therapeutic effects were observed in the mouse model of 5-FU-induced intestinal mucositis.

5-FU exerts antitumor effects by interfering with DNA synthesis in tumor cells, but it also interferes with DNA synthesis in normal tissue cells [21]. We previously demonstrated that 5-FU promotes cellular necrosis and the release of considerable amounts of proinflammatory double-stranded DNA, activates NLRP3 inflammasome, and induces an inflammatory response in mice [22]. NLRP3 inflammasome is a major member of the inflammasome family, which can be activated by bacterial toxins, ATP, reactive oxygen species, urea crystals and other pathogens and danger signal molecules in vivo; it is an important factor in anti-infection immunity and inducing inflammatory diseases [23]. Bauer et al. [24] found that NLRP3 inflammasome plays a role in gastrointestinal inflammatory diseases mainly by upregulating the expression of IL-1β, IL-18, and other inflammatory cytokines. In the absence of any intestinal infection, a low level of NLRP3 is expressed in intestinal mucosal epithelial cells and immune cells. During inflammation, the expression level of NLRP3 in immune cells increases rapidly, and NLRP3 further activates the protease caspase-1, which mediates the cleavage and maturation of IL-1β and IL-18 precursors, and eventually induces inflammatory reactions, leading to inflammatory injury in normal tissues [25]. In this study, both native Ber and Ber-CDs inhibited 5-FU-induced activation of NLRP3 and reduced the expression level of the proinflammatory cytokine IL-1β in mice. Furthermore, Ber-CD exhibited potent anti-inflammatory activities. Our findings suggest that Ber-CDs exert strong anti-inflammatory effects, possibly by inhibiting NLRP3 activation, reducing IL-1β expression level, and increasing IL-10 expression level. These compounds further protect normal intestinal mucosal tissues to maintain the integrity of the intestinal mucosal barrier.

The intestinal mucosal barrier is composed of mechanical, biological, immune and chemical barriers. The mechanical barrier comprises a mucus layer on the surface of the intestinal mucosa and tight junctions between closely arranged epithelial cells [26]. 5-FU can severely damage the villi and crypt structures in the small intestine, destroy the tight junctions between mucosal epithelial cells, and damage the intestinal mucosal barrier [27]. 5-FU inhibits the expression of the tight junction proteins occludin and ZO-1 in intestinal epithelial cells, thus increasing the permeability of the intestinal mucosa so that intestinal bacteria and bacterial toxins (e.g., endotoxin and lipopolysaccharides) can enter the bloodstream through the intestinal mucosa and cause systemic inflammation [28]. These changes eventually increase oxidative stress and damage normal tissues. After native Ber and Ber-CDs treatments, considerable increases were noted in the levels of endotoxin and oxidative stress response in the blood of the experimental mice. The condition of mice with CIM was significantly better than that of those with 5-FU-induced intestinal mucositis; furthermore, and the expression levels of ZO-1 and Occludin in intestinal epithelial cells were significantly higher in mice with CIM than in those with 5-FU-induced intestinal mucositis. Thus, Ber can promote the expression of ZO-1 and Occludin to maintain the integrity of mucosal tight junctions and prevent the intestinal bacteria entering the bloodstream. Our findings are consistent with those of other studies [29,30]. However, we found no significant differences in intestinal mucosal barrier integrity or oxidative response indices in mice treated with Ber-CDs and those treated with native Ber. This might be because of the short treatment duration; thus, further studies are necessary.

IgA secreted by intestinal lamina propria plasma cells and intestinal epithelial cells is also an important part of the intestinal mucosal barrier, as it is the first line of defense against pathogen invasion, adhesion, and colonization in the intestinal mucosa [31]. 5-FU markedly inhibited IgA secretion in the experimental mice, thus rendering the host susceptible to various pathogens [32]. We found that the plasma levels of IgA were considerably higher in mice treated with Ber-CDs than in those treated with 5-FU or Con-Ber. These findings indicated that Ber-CDs effectively enhanced mucosal immunity and prevented inflammation induced by intestinal bacteria or their endotoxins.

SCFAs are organic fatty acids with 1–6 carbon atoms. Some microorganisms residing in the human colon produce SCFAs to ferment dietary fibers and resistant starch that cannot be digested by humans [33]. Three major SCFAs found in the human colon are acetic, propionic, and butyric acids, which account for >95% of all SCFAs [34]. SCFAs can regulate host intestinal mucosal immunity, reduce the colonic inflammatory response, inhibit colon tumor cell proliferation, and induce tumor cell differentiation and apoptosis [35]. Through SCFAs and other metabolites, gut microbes regulate the changes in the physiological functions of their host [36]. SCFAs also participate in the metabolic activities of different organs of the human body and exert various effects. Acetic acid produced through bacterial fermentation can be absorbed and used by the host. It is a key source of host energy, providing approximately 10% of total daily energy [37]. After its absorption in the blood, propionate is catabolized in the liver, and it participates in the conversion of pyruvate to glucose and may inhibit the synthesis of fat [38]. Butyrate can be absorbed by epithelial cells and is the main source of energy for these cells [39]. Butyric acid exerts strong anti-inflammatory effects by activating G-protein-coupled receptors and inhibiting histone deacetylases [40]. In animal studies, butyrate inhibited the bacterial endotoxin–induced activation of neutrophils, the release of various proinflammatory cytokines (e.g., tumor necrosis factor-α), and various inflammation-related signaling pathways (e.g., nuclear factor-κB signaling) and reduced inflammatory reactions [41,42,43]. SCFAs help maintain the integrity of the intestinal mucosal barrier and increase the secretion of mucin by mucosal epithelial cells to enhance a host’s ability to resist attack by pathogens [44,45]. We found that the relative abundances of *Bifidobacterium* and *Lactobacillus* in the colon contents of the mice were significantly increased in the Ber-CD and Con-Ber groups. Further LC–MS/MS analysis revealed that the levels of acetic, propionic, and butyric acids in the colon contents of the experimental mice were significantly increased in the Ber-CD and Con-Ber groups; the levels of these three SCFAs were significantly higher in the Ber-CD group than in the Con-Ber group. CDs may improve the effectiveness of native Ber in vivo; however, further studies must be conducted to elucidate the exact mechanisms.

## 4. Materials and Methods

### 4.1. Ber-CD Synthesis and Identification

The nitrogen flow rate of a supersonic collision crushing instrument was set to 600 m/s. Ber was instantaneously crushed into ultrafine powder through super-high-speed collision twice. The temperature of a photoelectric carbonization instrument was set to 120 °C to heat the mixture of Ber powder and water. The mass ratio of Ber powder and water was 1:10 (m/m), the heating time was 180 s, and the far-infrared irradiation speed was 30/s, further give the resulted solution a hydrothermal treatment for 3 h under 180 °C to ensure the uniform dispersion of carbon nanoparticles in water to form a colloidal solution of CDs. The morphology and particle size of Ber-CDs were assessed through transmission electron microscopy (JEM-2100Plus, JEOL, Kitakyushu, Japan). The fluorescence emission and excitation spectra of Ber-CDs were obtained through fluorescence spectrometry (Infinite E Plex, TECAN, Männedorf, Switzerland).

### 4.2. Animal Experiment

Male specific pathogen–free C57BL/6 mice (*n* = 32; age, 8 weeks; body weight, 18–22 g) were purchased from the Animal Center of Yangzhou University (Yangzhou, China) and housed in the Animal Experimental Center of Jiangsu University, China. Mice were randomly divided into the following four groups: NC, 5-FU, Ber-CD, and Con-Ber groups. Each group comprised eight mice. The NC group was fed normally and received no treatment. The mice in the 5-FU, Ber-CD, and Con-Ber groups were intraperitoneally injected with 5-FU (30 mg/kg, King York Inc., Tianjin, China) for 5 days to induce intestinal mucositis. The mice in the Ber-CD and Con-Ber groups were intraperitoneally injected with Ber-CDs or native Ber (5 mg/kg), respectively, every day from Day 3. The experiment ended on Day 5. Then, serum, spleen, and intestinal tissues were collected from all groups for subsequent analyses.

### 4.3. qRT-PCR Assay

Ttotal RNA was extracted from the spleen tissues of the experimental mice by following the standard Trizol method (Vazyme, Nanjing, China). cDNA was synthesized through reverse transcription with oligo(dT)_n_ as primer. The mRNA expressions of inflammatory related factors in the spleen were determined by fluorescent quantitative PCR (Vazyme, Nanjing, China). The total system of qRT-PCR reaction was 20 μL, including 10 μL SYBR Green Master Mix, 0.4 μL (10 μmol/L) of upper and lower primers, and 2 μL cDNA template. The following primer sequences were used: *NLRP3* (Forward: 5′-ATTACCCGCCCGAGAAAGG-3′, Reverse: 5′-CATGAGTGTGGCTAGATCCAAG-3′), *IL-10* (Forward: 5′-GAAGCTCCCTCAGCGAGGACA-3′, Reverse: 5′-TTGGGCCAGTGAGTGAAAGGG-3′) and *β-actin* (Forward: 5′-ATGACCCAAGCCGAGAAGG-3′, Reverse: 5′-CGGCCAAGTCTTAGAGTTGTTG-3′). The reaction procedure was as follows: pre-denaturation at 95 °C for 5 min, denaturation at 95 °C for 3 s, annealing at 58 °C for 20 s, and extension at 72 °C for 30 s. There were 40 cycles in total. With GAPDH as the internal reference, the relative mRNA expression was calculated by using 2^−(ΔΔCT)^.

### 4.4. ELISA Assay

ELISA assay was used to detect the expressions of IL-1β and IgA in the serum of experimental mice, and enzyme-labeled instrument was used to detect the absorbance value at 450 nm. The standard curves were used to calculated the concentrations of IL-1β and IgA.

### 4.5. HE and Immunohistochemical Staining

Proximal ileocecal intestinal tissues were fixed in 4% paraformaldehyde for 48 h, embedded in paraffin, and stained with HE. The integrity of the intestinal mucosal villus epithelium, separation of the mucosa and lamina propria, edema of the muscle layer, and shedding of intestinal mucosal villi were observed under a microscope.

Paraffin-embedded tissue samples were subjected to immunohistochemical staining. After dewaxing, gradient dehydration, antigen repair, blocking with goat serum, and incubation with antibodies, the tissues samples were stained with 3,3′-diaminobenzidine and counterstained with hematoxylin. The slices were sealed with neutral resin and observed under a microscope. Image-Pro Plus (version 6.0) (Media Cybernetics, Rockville, MD, USA) was used to measure the integral optical density to evaluate the expression levels of occludin and ZO-1.

### 4.6. Measurement of Relative Abundances of Intestinal Bacteria and Levels of SCFAs in Mouse Feces

qRT-PCR was performed to evaluate the relative abundances of intestinal bacteria. For this, primers were synthesized by Genewiz (Suzhou, China). For the investigation, we selected four key intestinal bacteria, namely *Bifidobacterium*, *Lactobacillus*, *E. coli*, and *Enterococcus*. In this study, the colon contents of the experimental mice were collected. Total DNA was extracted using the Fecal Genomic DNA extraction kit (Tiangen Biotech, Beijing, China). The PCR products of the aforementioned four bacteria were ligated to the pTG19-T vector (Generay Biotech, Shanghai, China) to prepare the gene diluents with known copy number. A series of diluents (10^2^ copy/mL to 10^11^ copy/mL) were used to generate the amplification standard curve. The copy numbers of the target bacteria in the samples were calculated according to the standard curve.

The levels of the three main SCFAs detected in mouse feces were measured by Microeco Tech Co., Ltd. (Shenzhen, China) through gas chromatography (GC)–MS. In brief, 50 mg of fecal sample was mixed with 500 mL of saturated NaCl solution; next, 20 mL of 10% sulfuric acid solution was added for acidification, which was followed by the addition of 800 mL of diethyl ether before shaking. The mixture was centrifuged at 18,000× *g* for 15 min at 4 °C. The supernatant was collected, and 0.25 g anhydrous sodium sulfate was added to it; the mixture was shaken and centrifuged again. The supernatant was obtained and added to a gas phase flask for GC–MS analysis.

### 4.7. Detection of Serum Endotoxin, SOD and MDA

Serum endotoxin was detected by the medical laboratory department of Northern Jiangsu People’s Hospital (Yangzhou, China); for this, the Tachypleus amebocyte lysate method was used. The activity of SOD in mouse serum was evaluated using a xanthine oxidase assay kit (Jiancheng Bioengineering Institute, Nanjing, China). Furthermore, the activity of MDA in mouse serum was evaluated using a thiobarbituric acid assay kit (Jiancheng Bioengineering Institute).

### 4.8. Statistical Analysis

Image J and GraphPad Prism8.0 analysis software were used for image analysis and data processing. SPSS 22.0 software was used for statistical analysis. All data were expressed as means ± standard deviations (SD). One-way ANOVA was used for comparison between multiple groups, and LSD-T test was used for statistical analysis for multiple comparisons between groups. *p* < 0.05 indicates statistical significance.

## 5. Conclusions

Ber-CDs may alleviate inflammation and maintain intestinal mucosal immunity in vivo. The underlying mechanisms may involve the alteration of intestinal flora to increase the levels of SCFAs. Ber-CDs may alleviate inflammation by inhibiting the activation of intestinal mucosal immune cells and the production of various inflammatory factors. In addition, these compounds increase the levels of ZO-1 and occludin (tight junction proteins found in the intestinal mucosal barrier), thus improving the integrity of the mucosal barrier and preventing the proliferation of intestinal bacteria (Figure 7).

## Figures and Tables

**Figure 1 molecules-28-02148-f001:**
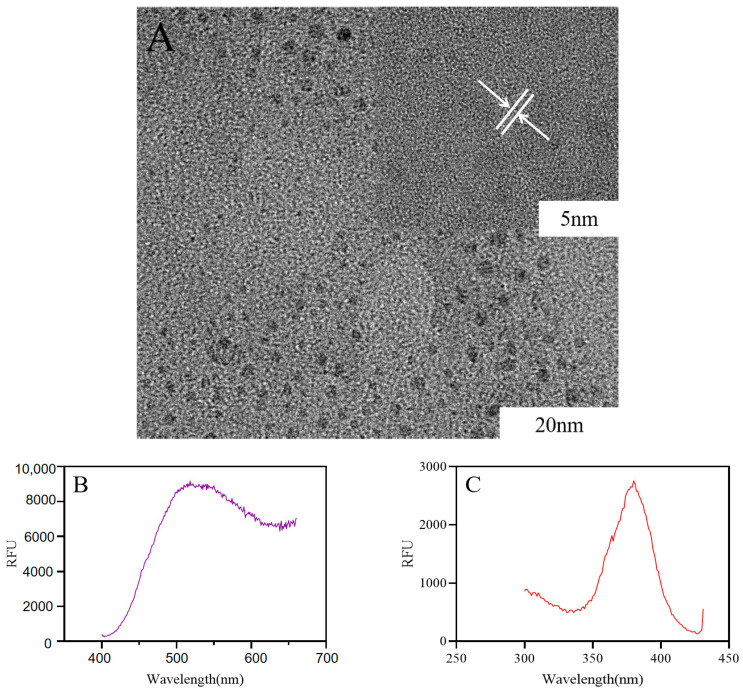
The structure of Ber-CDs observed by transmission electron microscope. (**A**) Transmission electron microscope; (**B**) Ber-CDs emission wavelength; (**C**) Ber-CDs excitation wavelength.

**Figure 2 molecules-28-02148-f002:**
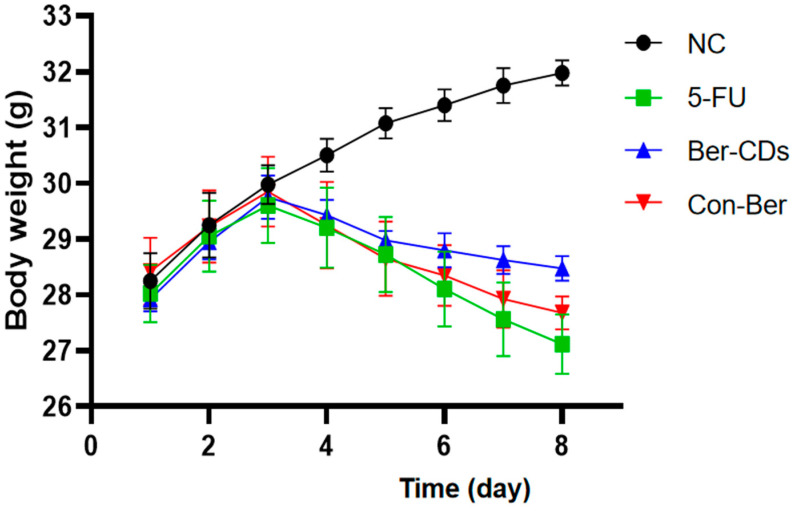
Changes of body weight in each group during the experiment. C57BL/6 mice (*n* = 8/each group) in the 5-FU, Ber-CDs and Con-Ber groups were intraperitoneally injected with 5-FU (30 mg/kg) to induce chemotherapy-induced intestinal mucositis. Mice in the Ber-CDs and Con-Ber groups were intraperitoneally injected with Ber-CDs or Ber (5 mg/kg) for 3 days, respectively.

**Figure 3 molecules-28-02148-f003:**
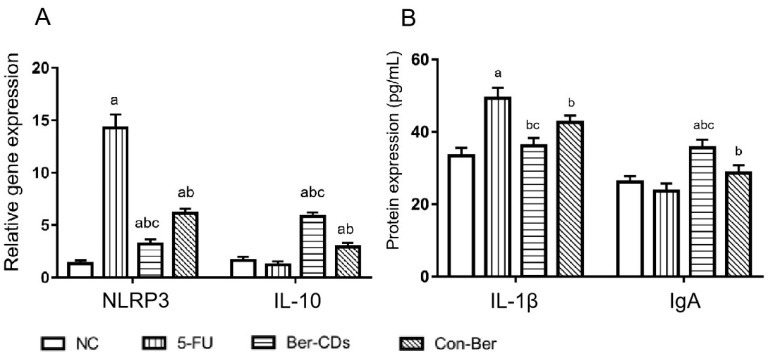
Expressions of NLRP3, IL-10, IL-1β, IgA detected by qPCR and ELISA assay. (**A**) Relative gene expression of NLRP3 and IL-10 in the spleen; (**B**) Protein levels of IL-1β and IgA in the serum; a: compared with the NC group, *p* < 0.05; b: compared with the 5-FU group, *p* < 0.05; c: compared with the Con-Ber group, *p* < 0.05.

**Figure 4 molecules-28-02148-f004:**
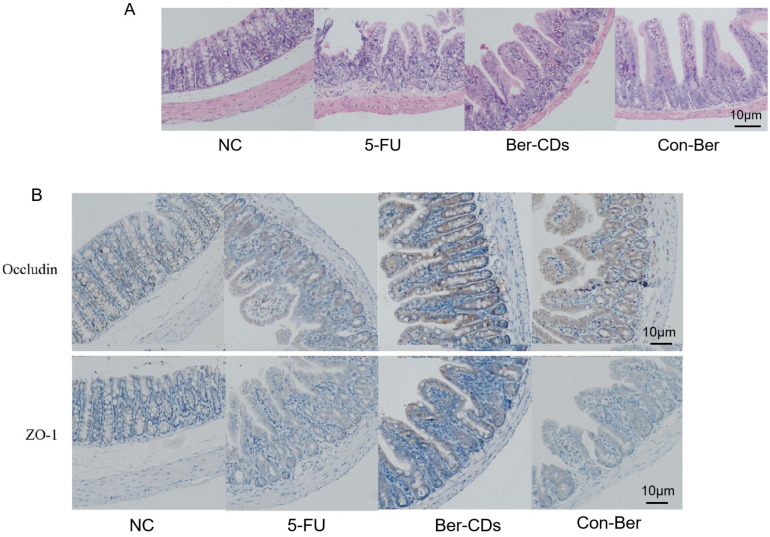
HE staining and immunohistochemistry were used to detect intestinal tissue of mice. (**A**) HE staining of mouse intestinal tissue; (**B**) The expressions of Occludin and ZO-1 in intestinal tissues of mice were detected by immunohistochemistry. b: compared with 5-FU, *p* < 0.05.

**Figure 5 molecules-28-02148-f005:**
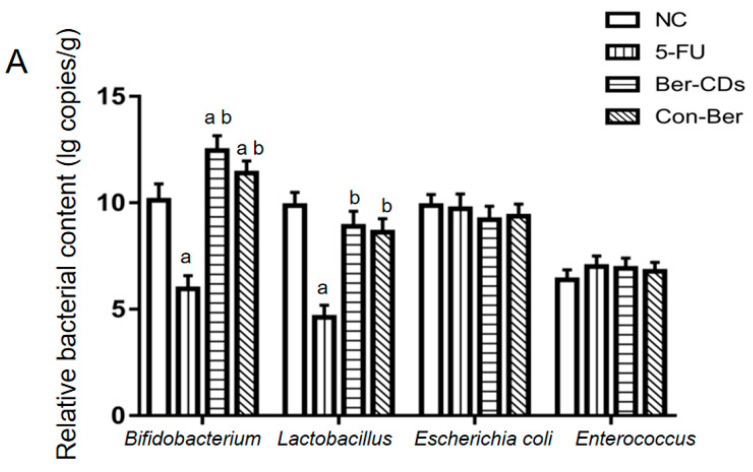
The relative contents of four important bacteria and the concentrations of three main SCFAs in colon contents of mice. (**A**) Reletive bacterial contents. (**B**) SCFAs levels. a: compared with the NC group, *p* < 0.05; b: compared with the 5-FU group, *p* < 0.05; c: compared with the Con-Ber group, *p* < 0.05.

**Figure 6 molecules-28-02148-f006:**
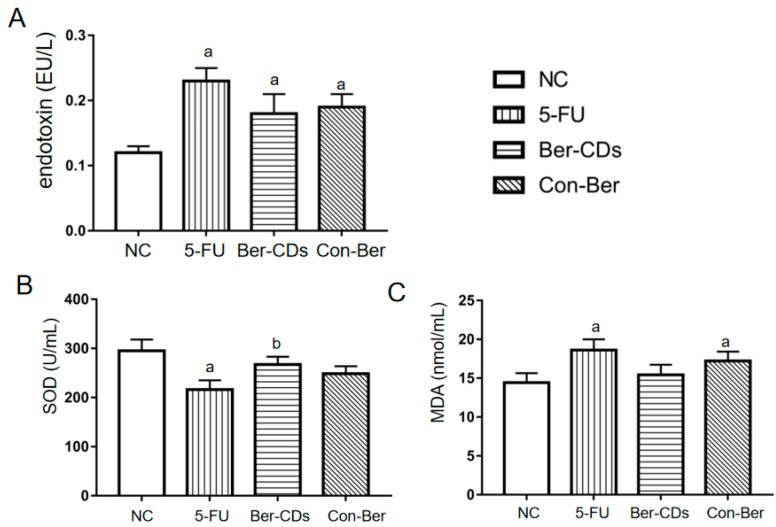
The concentrations of endotoxin, SOD and MDA in mouse plasma. (**A**) Plasma endotoxin levels. (**B**) Plasma SOD levels. (**C**) Plasma MDA levels. a: compared with the NC group, *p* < 0.05; b: compared with the 5-FU group, *p* < 0.05.

**Figure 7 molecules-28-02148-f007:**
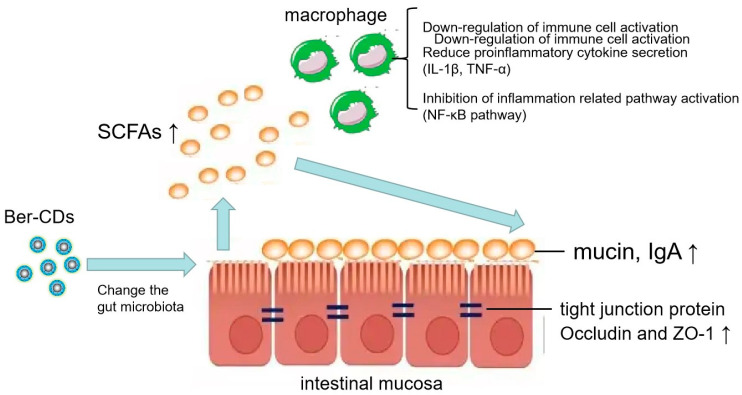
Schematic illustration of mechanism. Schematic illustration of the effects of berberine-based carbon quantum dots on 5-FU-induced intestinal mucositis in C57BL/6 mice via enhancing gut-derived short chain fatty acids contents and attenuating intestinal barrier injury and oxidative stress.

## Data Availability

The data presented in this study are available on request from the corresponding authors.

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
