# Peer review of "Berberine-Based Carbon Quantum Dots Improve Intestinal Barrier Injury and Alleviate Oxidative Stress in C57BL/6 Mice with 5-Fluorouracil-Induced Intestinal Mucositis by Enhancing Gut-Derived Short-Chain Fatty Acids Contents"

_molecules, 2023, doi:10.3390/molecules28052148_

Round 1
Reviewer 1 Report
Comments to the Author
In this manuscript, the authors synthesized a berberine-based carbon quantum dots (Ber-CDs) and mainly evaluated its improvement effect on 5-fluorouracil-induced intestinal mucositis in C57BL/6 mice, and explored the mechanism behind this effect through a variety of tests. The effect of protecting intestinal mucosal immunity is favorable, and has certain practical significance. However, in some sections, there are still some shortcomings that require more in-depth discussion and a more detailed explanation of the results.
1. The language of the manuscript needs to be further refined to enhance the logic of the text, especially in the introduction and discussion sections. Also, the preciseness of the words used in results section should be improved.
2. There are too many errors in the manuscript to list them all.
(1) “Space errors” are extremely serious. There is no need to separate a space between the number and °C or %, but spaces between numbers and nm.
(2) It is necessary to ensure a consistent format when describing images, such as replacing “Figure 4-A” with “Figure 4A”. Also, the letters in Figure 1 should be changed from lowercase to uppercase. Besides, the drawings need to be improved.
(3) The unit of solution volume in line 336 should be unified as “μL”.
(4) The authors need to carefully examine the entire manuscript verbatim.
3. According to Figure 6B, SOD concentration in the 5-FU group should have decreased, and the description in line 188 is incorrect. And it is recommended to indicate in this paragraph that this section is a description of Figure 6.
4. The fluorescence stability of CDs is mentioned in the introduction and discussion sections, and it is suggested to supplement the fluorescence stability experiments of Ber-CDs for illustration.
5. This manuscript deals less with the experiment itself. What are the effects of experimental temperature and time on the morphology and performance of synthesized Ber-CDs? Whether relevant experiments can be supplemented to illustrate?
6. To enhance the comprehensiveness, these important and closely related literatures should be cited: Mater. Today., 2022, 54, 42-51.; Molecules, 2023, 28, 1451.; J Colloid Interface Sci, 2023, 637, 173-181.
Author Response
Reviewer 1#
In this manuscript, the authors synthesized a berberine-based carbon quantum dots (Ber-CDs) and mainly evaluated its improvement effect on 5-fluorouracil-induced intestinal mucositis in C57BL/6 mice, and explored the mechanism behind this effect through a variety of tests. The effect of protecting intestinal mucosal immunity is favorable, and has certain practical significance. However, in some sections, there are still some shortcomings that require more in-depth discussion and a more detailed explanation of the results.
Dear Reviewer,
Thank you for your rapid response on our manuscript. The constructive criticism of you was much appreciated and we revised our manuscript accordingly. The suggestions were accepted, more information and details were included in the text, and the manuscript was revised thoroughly. On the other hand, we rearranged the manuscript to improve the quality. All the modifications performed in the revised manuscript are highlighted in red. The enclosed document at the bottom of this letter contains a point-to-point reply to you comments.
- The language of the manuscript needs to be further refined to enhance the logic of the text, especially in the introduction and discussion sections. Also, the preciseness of the words used in results section should be improved.
Response: The revised manuscript has been copyedited by a language editor who is a native English speaker with extensive scientific/technical background (No. OD-2023-001529-01, www.wallaceediting.cn).
- There are too many errors in the manuscript to list them all.
(1) “Space errors” are extremely serious. There is no need to separate a space between the number and °C or %, but spaces between numbers and nm.
Response: Thank you for your comment. We checked and revised all similar problems in the manuscript.
(2) It is necessary to ensure a consistent format when describing images, such as replacing “Figure 4-A” with “Figure 4A”. Also, the letters in Figure 1 should be changed from lowercase to uppercase. Besides, the drawings need to be improved.
Response: Thank you for your comment. We revised.
(3) The unit of solution volume in line 336 should be unified as “μL”.
Response: Thank you for your comment. We revised.
(4) The authors need to carefully examine the entire manuscript verbatim.
Response: We checked the revised manuscriptcarefully.
- According to Figure 6B, SOD concentration in the 5-FU group should have decreased, and the description in line 188 is incorrect. And it is recommended to indicate in this paragraph that this section is a description of Figure 6.
Response: Thank you for your comment. We revised.
- The fluorescence stability of CDs is mentioned in the introduction and discussion sections, and it is suggested to supplement the fluorescence stability experiments of Ber-CDs for illustration.
Response: Thanks for the comments. Generally, the fluorescence intensity of CDs is stable, we added the stability property of Ber-CDs assay in the revised manuscript. As shown in supplementary Figure 1 the fluorescence intensity of the Ber-CDs is relatively stable even after 6 months.
- This manuscript deals less with the experiment itself. What are the effects of experimental temperature and time on the morphology and performance of synthesized Ber-CDs? Whether relevant experiments can be supplemented to illustrate?
Response: The purpose of this manuscript was to investigate the best conditions of synthesis for high quality Ber-CDs for biological applications, specifically the temperature and time required for their synthesis. Therefore, the synthesized Ber-CDs we use are prepared under the optimal conditions that based on previous reports (J. Mater. Chem., 2012,22, 24230-24253; Carbon, 2011,49, 605-609; and Mater. Lett., 2019, 238, 22-25 ). We also conducted relevant experiments, and found that the synthesized Ber-CDs still has good stable performance after 6 months as shown in Figure S1.
- To enhance the comprehensiveness, these important and closely related literatures should be cited: Mater. Today., 2022, 54, 42-51.; Molecules, 2023, 28, 1451.; J Colloid Interface Sci, 2023, 637, 173-181.
Response: Thank you for your comment. We have added these important references in appropriate places.

Reviewer 2 Report
The authors report on the administration of carbon based quantum dots doped with berberine to elevate the detrimental effect of 5-fluorouracil, an aggressive cytotoxic cancer drug. In particular, the intestines are severely affected by treatment of 5-fluorouracil. The authors show in a mouse study that their quantum dots to alleviate some of the issues caused by the cancer drug, with reduced weightless of the mice, reduced inflammation markers, better inflammation morphology of the intestines, healthier gut flora and reduced endotoxins. In most of the cases, berberine, not in the quantum dot, also showed similar results. It would had liked to see better improvement with the treatment. Nevertheless, it is an improvement and can undoubtedly be improved.
The manuscript is well written and presented in a logical manner.
I am happy to recommend this manuscript for publication.
Author Response
Reviewer 2#
The authors report on the administration of carbon based quantum dots doped with berberine to elevate the detrimental effect of 5-fluorouracil, an aggressive cytotoxic cancer drug. In particular, the intestines are severely affected by treatment of 5-fluorouracil. The authors show in a mouse study that their quantum dots to alleviate some of the issues caused by the cancer drug, with reduced weightless of the mice, reduced inflammation markers, better inflammation morphology of the intestines, healthier gut flora and reduced endotoxins. In most of the cases, berberine, not in the quantum dot, also showed similar results. It would had liked to see better improvement with the treatment. Nevertheless, it is an improvement and can undoubtedly be improved. The manuscript is well written and presented in a logical manner. I am happy to recommend this manuscript for publication.
Response: Dear Reviewer, thank you for your rapid response on our manuscript.
Round 2
Reviewer 1 Report
Accepted.